# VARICOT: A VARIATIONAL FRAMWORK FOR IMPLICIT REASONING AS STRUCTURED LATENT INFERENCE

## ABSTRACT

Chain-of-Thought (CoT) elicits remarkable capabilities in large language models but is fundamentally constrained by the low-bandwidth, sequential nature of text generation. Implicit CoT methods promise to accelerate by reasoning on latent space, yet they often rely on heuristic architectures and complex multi-stage training, lacking a unified, principled foundation. We introduce VARICOT, the first principled variational framework that formulates implicit reasoning as a structured probabilistic inference problem. VARICOT learns a continuous latent variable, Z, that represents the entire reasoning process, optimized via a single, unified evidence lower bound (ELBO) objective. Our key architectural innovation, Guided Latent Reasoning, treats Z as a global reasoning context that modulates the model's computations at every layer via cross-attention. This design decouples the abstract reasoning state from the linguistic realization, enabling high-bandwidth guidance without altering the standard autoregressive generation process. Implemented within a single Transformer and trained end-to-end with strategic control tokens, VARICOT offers flexible inference: either generating answers directly for a >2.5x speedup or reproducing the full rationale when needed. On benchmarks like GSM8K and CommonsenseQA, VARICOT substantially improves upon or matches the accuracy of explicit CoT while drastically reducing latency, establishing a theoretically grounded and scalable paradigm for efficient reasoning.

## 1  INTRODUCTION

Large language models (LLMs) have demonstrated remarkable reasoning capabilities, particularly when guided by explicit chain-of-thought (CoT) prompting that verbalizes intermediate steps in natural language (Wei et al., 2022; Kojima et al., 2022). While effective, this paradigm imposes a fundamental bottleneck: models capable of manipulating thousands of dimensions in their internal hidden states are forced to reason through the narrow, discrete channel of token-by-token generation. (Zhu et al., 2025). This mismatch between the model's high-dimensional computational capacity and the low-bandwidth nature of text couples the act of "thinking" with the act of "writing," incurring significant computational overhead and hindering end-to-end optimization of the reasoning process itself.

The limitations of explicit CoT have motivated a shift toward *latent reasoning*—performing multi-step inference entirely within the model's continuous hidden space without generating intermediate tokens. Large language models (LLMs) can solve complex problems by externalizing their reasoning process as a "chain of thought" (CoT) in natural language. However, this paradigm introduces a fundamental paradox: models capable of manipulating thousands of dimensions in their internal hidden states are forced to reason through the narrow, discrete channel of token-by-token generation.

The limitations of explicit CoT have spurred the development of latent reasoning—performing multi-step inference within the model's continuous hidden space without generating intermediate tokens (Geiping et al., 2025; Ruan et al., 2025; Hao et al., 2024; Shen et al., 2025). While promising, current approaches represent a zoo of disparate, often heuristic solutions. They range from injecting "latent tokens" into the input sequence (vertical reasoning) to modifying hidden states across layers (horizontal reasoning) (Zhu et al., 2025). More critically, these methods frequently depend on complex, multi-stage training pipelines, such as distillation from an explicit CoT "teacher" model or reliance on external memory modules (Dehghani et al., 2018; Sun et al., 2024; Behrouz et al.,

2024; Shen et al., 2025). This architectural fragmentation and lack of a shared theoretical objective prevent true end-to-end optimization, making the learned reasoning processes opaque and difficult to generalize. What is missing is a unified framework that combines the efficiency of latent reasoning with a principled learning objective.

In this work, we introduce **VARICOT (Variational Implicit Chain-of-Thought)**, a unified framework that addresses these challenges by reformulating implicit reasoning as a problem of structured probabilistic inference. Instead of heuristics, we treat the unobserved reasoning process as a continuous, structured latent variable Z. We then derive a single evidence lower bound (ELBO) objective that jointly learns to: (1) infer Z from the problem context, (2) generate the final answer conditioned on Z, and (3) optionally reconstruct the explicit reasoning trace for interpretability. This principled, variational foundation allows for end-to-end training of the entire reasoning architecture within a single model.

VARICOT is realized through two synergistic innovations. First, we propose **Guided Latent Reasoning**, a novel architectural paradigm that cleanly decouples the latent reasoning state (Z) from the token-level representations. In our design, Z acts as an external, global context that is shared across all Transformer layers. At each layer, the model uses cross-attention to query this latent context, allowing the reasoning state to guide linguistic processing without being entangled in the residual stream. This synthesizes the representational power of continuous states with the structural integrity of autoregressive models. Second, we employ **Strategic Control Tokens** Goyal et al. (2024); Wang et al. (2024) (e.g., `<prior>`, `<posterior>`) to manage the different probabilistic operations (prior sampling, posterior inference, and conditional generation) within a single, standard autoregressive pass. This lightweight mechanism eliminates the need for multi-stage pipelines or architectural modifications, enabling seamless, scalable, and end-to-end training.

We evaluate variCoT across arithmetic, symbolic, and commonsense reasoning benchmarks. Our framework consistently outperforms strong explicit CoT and latent baselines, while exhibiting superior sample efficiency and robustness to prompt perturbations. Ablation studies confirm that the variational objective is essential: it not only improves performance but also encourages disentangled, interpretable latent representations that align with ground-truth reasoning steps.

In summary, our contributions are:

- **variCoT, A Unified Variational Framework**: We are the first to formalize implicit CoT reasoning within a principled variational inference framework, optimizing a joint evidence lower bound (ELBO) in an end-to-end fashion.

- **Guided Latent Reasoning**: We introduce a novel architecture that decouples reasoning and language by using the latent reasoning state as a cross-attentional query, providing high-bandwidth guidance to every layer of the Transformer.

- **Single-Model, End-to-End Training**: We demonstrate how strategic control tokens can embed the entire variational machinery into a standard Transformer, eliminating the need for complex, multi-stage training pipelines like distillation.

- **Improved Performance**: We will show through extensive experiments on arithmetic, commonsense, and symbolic reasoning tasks that VARICOT achieves state-of-the-art accuracy and sample efficiency while offering significant inference speedups (>2.5x) over explicit CoT methods.

## 2 METHODOLOGY

We introduce variCoT, a unified variational framework for implicit Chain-of-Thought reasoning that addresses fundamental limitations in existing latent reasoning approaches. While methods like explicit CoT are constrained by discrete token sequences and latent approaches often rely on heuristic architectures or multi-stage training, variCoT provides a principled probabilistic foundation for learning continuous reasoning traces within a single Transformer. This section formalizes our approach through a structured generative model, derives its training objective via variational inference, and demonstrates how each component overcomes key challenges in latent reasoning.

## 2.1 BACKGROUND AND NOTATION

We begin by establishing the formal setting for reasoning in large language models. Let $X^q = (x_1^q, ..., x_n^q)$ denote the input question token sequence, $Y^r = (y_1^r, ..., y_m^r)$ the explicit reasoning chain, and $Y^a = (y_1^a, ..., y_k^a)$ the final answer. Standard autoregressive language models generate these components sequentially using the factorization $p(Y^r, Y^a \mid X^q) = p(Y^r \mid X^q) \cdot p(Y^a \mid X^q, Y^r)$.

The fundamental limitation of this approach lies in the information bottleneck of discrete tokens. Each token carries approximately 15 bits of information, while a single hidden state in modern LLMs (e.g., 4096-dimensional) can encode 40,960 bits—a 2,700× increase in expressive capacity Zhu et al. (2025). This observation has motivated latent reasoning methods that operate in continuous hidden spaces. However, existing approaches such as Coconut Hao et al. (2024) and CODI Shen et al. (2025) rely on deterministic recurrence or distillation pipelines, lacking proper uncertainty quantification and end-to-end optimization.

variCoT addresses these limitations by introducing a sequence of continuous latent variables $Z = (z_1, ..., z_L)$ that serves as a compressed, stochastic representation of the reasoning process. Unlike prior work, our framework formalizes $Z$ within a generative model, enabling principled variational inference and uncertainty-aware reasoning while maintaining full compatibility with standard Transformer architectures.

## 2.2 THE VARICOT FRAMEWORK

variCoT is grounded in two key insights from the latent reasoning literature: (1) the expressive advantage of continuous hidden states over discrete tokens, and (2) the functional specialization of Transformer layers—shallow layers for representation, intermediate for transformation, and deep for integration Skean et al. (2024); Gromov et al. (2024); Shi et al. (2024); Zhang et al. (2024). We mirror this structure by letting $Z$ encapsulate the full reasoning trajectory before branching into separate decoders for reasoning and answer generation. We begin by establishing a general theoretical foundation for variational reasoning without imposing any assumption:

**Theorem 2.1** (Evidence Lower Bound for Latent Reasoning). *For any joint distribution $p(Y^r, Y^a, Z \mid X^q)$ and variational approximation $q_\phi(Z \mid X^q, Y^r, Y^a)$, the log marginal likelihood admits the decomposition:*

$$\log p(Y^r, Y^a \mid X^q) = \mathcal{L}_{ELBO} + D_{\mathrm{KL}}\left(q_\phi(Z \mid X^q, Y^r, Y^a) \,\|\, p(Z \mid X^q, Y^r, Y^a)\right),$$

*where*

$$\mathcal{L}_{ELBO} = \mathbb{E}_{q_\phi}\left[\log \frac{p(Y^r, Y^a, Z \mid X^q)}{q_\phi(Z \mid X^q, Y^r, Y^a)}\right].$$

*Proof.* The derivation follows from a variational decomposition of the log marginal likelihood, leveraging the non-negativity of the Kullback-Leibler divergence. A complete derivation is provided in Appendix A.1. □

While Theorem 2.1 provides a general variational foundation, it presents two practical challenges for reasoning applications. First, the KL divergence term requires access to the true posterior $p(Z \mid X^q, Y^r, Y^a)$, which is intractable. Second, even with a variational approximation $q_\phi$, the posterior remains conditioned on both $Y^r$ and $Y^a$, making it unusable during inference when reasoning chains are unavailable.

To address these limitations, we introduce a structured generative model that enables tractable optimization and practical deployment. Our approach is motivated by the observation that effective reasoning requires a clean separation between abstract computation and linguistic realization.

**Assumption 2.2** (Latent Reasoning Mediation). *There exists a sequence of latent reasoning states $Z = (z_1, \ldots, z_L)$ such that, conditioned on the question $X^q$ and $Z$, the explicit reasoning $Y^r$ and the answer $Y^a$ are conditionally independent:*

$$Y^r \perp\!\!\!\perp Y^a \mid X^q, Z.$$

This assumption reflects the cognitive intuition that once the core reasoning process is complete, its verbalization ($Y^r$) and final answer ($Y^a$) can be generated independently. It aligns with empirical

findings on layer-wise specialization in Transformers, where shallow layers handle surface features while deeper layers integrate semantic and inferential content.

Under Assumption 2.2, we obtain a tractable factorization of the joint distribution:

**Proposition 2.3** (variCoT Generative Factorization). *Under Assumption 2.2, the joint distribution over $Y^r$, $Y^a$, and $Z$ given $X^q$ factorizes as:*

$$p_{\theta,\psi,\rho}(Y^r, Y^a, Z \mid X^q) = p_\psi(Y^r \mid X^q, Z) \cdot p_\rho(Y^a \mid X^q, Z) \cdot p_\theta(Z \mid X^q),$$

*where $p_\theta(Z \mid X^q)$ is the prior over latent reasoning, and $p_\psi$, $p_\rho$ model the generation of explicit reasoning and answer, respectively.*

This factorization enables a computationally efficient training objective that bridges the theoretical ELBO with practical optimization:

**Theorem 2.4** (VariCOT Objective Decomposition). *Under the factorization in Proposition 2.3, the ELBO decomposes into three interpretable components:*

$$\mathcal{L}_{ELBO} = \underbrace{\mathbb{E}_{q_\phi}\left[\log p_\psi(Y^r \mid X^q, Z)\right]}_{\mathcal{L}_{reasoning}} + \underbrace{\mathbb{E}_{q_\phi}\left[\log p_\rho(Y^a \mid X^q, Z)\right]}_{\mathcal{L}_{answer}}$$
$$- \underbrace{\beta \cdot D_{\mathrm{KL}}\left(q_\phi(Z \mid X^q, Y^r, Y^a) \,\|\, p_\theta(Z \mid X^q)\right)}_{\mathcal{L}_{KL}},$$

*where $\beta > 0$ is a tunable regularization coefficient.*

*Proof.* The decomposition follows from substituting the structured joint distribution into the ELBO and applying linearity of expectation. See Appendix A.2. □

The decomposition in Theorem 2.4 provides a principled training objective where each term serves a distinct function. During training, the variational posterior $q_\phi(Z \mid X^q, Y^r, Y^a)$ absorbs all available information from both reasoning chains and answers. The KL regularization term $\mathcal{L}_{\mathrm{KL}}$ ensures that the prior $p_\theta(Z \mid X^q)$ learns to approximate this informed distribution, enabling effective inference when ground-truth reasoning chains are unavailable. This design allows the model to sample $Z \sim p_\theta(Z \mid X^q)$ at test time and generate $Y^a$ directly, enabling efficient latent-only reasoning that bypasses explicit CoT generation while retaining the ability to reconstruct rationales when interpretability is required. The remaining terms provide complementary learning signals: $\mathcal{L}_{\mathrm{reasoning}}$ ensures the latent variable $Z$ retains sufficient information to reconstruct explicit reasoning chains, serving as an interpretability anchor, while $\mathcal{L}_{\mathrm{answer}}$ drives task performance by ensuring $Z$ encodes all necessary information for accurate final answers.

This formulation establishes variCoT as a probabilistically grounded framework for end-to-end trainable latent reasoning. Compared to heuristic or distillation-based approaches, our method provides theoretical guarantees through its ELBO foundation while addressing key limitations of prior work: it enables uncertainty-aware reasoning through distributional latent states, supports generalization via prior regularization, and maintains architectural flexibility through modular decoders.

## 3 IMPLEMENTING VARICOT: THE GUIDED LATENT TRANSFORMER

The variCoT framework proposes a unified variational objective for latent reasoning. To realize its full potential, we must address two practical challenges: (1) how to train all components—prior, posterior, reasoning decoder, and answer decoder—efficiently within a single model, and (2) how to represent and inject the latent variable $Z$ to achieve high-bandwidth reasoning while maintaining architectural compatibility. We solve the first challenge through strategic control tokens that enable end-to-end training, and the second through guided latent reasoning, a novel architectural paradigm that synthesizes the strengths of existing approaches. The complete training and inference procedures are summarized in Algorithms 1 and 2 in the appendix.

## 3.1 Strategic Control Tokens: End-to-End Single-Model Training

A major limitation of existing latent reasoning frameworks is their reliance on multi-stage pipelines (Hao et al., 2024)—such as knowledge distillation (Shen et al., 2025), external encoders for discretization (Su et al., 2025), or persistent memory modules (Gao et al., 2024)—which fragment the computational graph, increase memory overhead, and hinder scalability within standard autoregressive architectures. We address this by introducing *strategic control tokens* that enable end-to-end variational inference within a single Transformer.

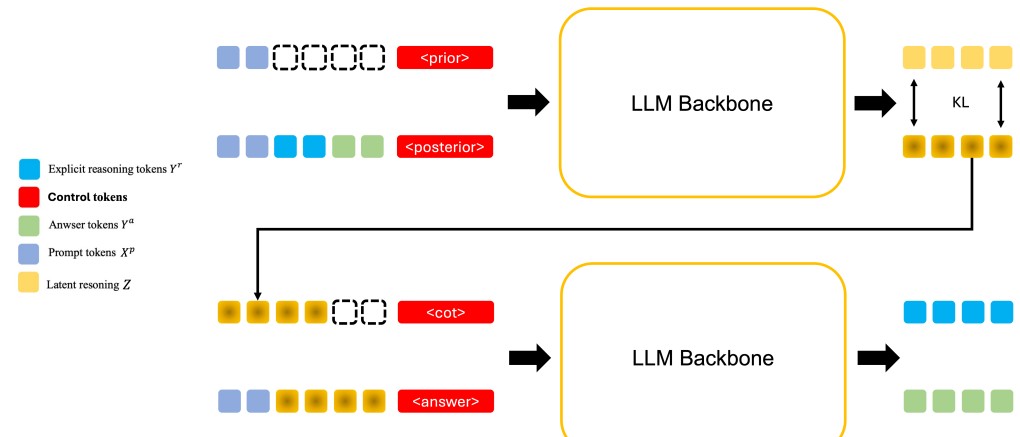

Figure 1: Training data flow in `variCoT`. Control tokens condition distinct probabilistic operations within a single forward pass. The latent variable $Z$ is sampled from the posterior and used to guide decoding. All components share parameters, enabling end-to-end training.

Our approach builds on training-induced recurrence (Goyal et al., 2024; Wang et al., 2024), where structured token sequences induce specialized computational roles without architectural modification. We extend this idea to variational learning by embedding the full generative and inference machinery into a unified sequence via functionally specialized tokens.

During training (Figure 1), a single forward pass processes the input question $X_q$, ground-truth reasoning trace $Y_r$, answer $Y_a$, and control tokens `<prior>` and `<posterior>`. The hidden state at `<posterior>` parameterizes the approximate posterior $q_\phi(Z \mid X_q, Y_r, Y_a)$, from which $Z$ is sampled and routed to the reasoning and answer decoders. Crucially, all components share the same Transformer parameters, enabling uninterrupted gradient flow. At inference, the model samples $Z$ from the prior $p_\theta(Z \mid X_q)$ and generates outputs autoregressively. For complete implementation details including token specifications and training protocols, see Appendix A.3.

By unifying probabilistic operations through token-level control, our method achieves full compatibility with pretrained LLMs while supporting expressive, uncertainty-aware reasoning—resolving key scalability and modularity challenges identified in recent latent reasoning literature (Sui et al., 2025).

## 3.2 Latent Representation Paradigms: Vertical, Horizontal, and Hybrid Approaches

Following the taxonomy of latent reasoning frameworks Zhu et al. (2025), we formalize three paradigms for representing the latent variable $Z$ in variational reasoning. Each defines a distinct architectural pathway for coupling latent reasoning states with autoregressive language modeling.

**Vertical Paradigm: Discrete Latent Tokens** In the vertical paradigm, the latent variable is instantiated as a sequence of discrete tokens $Z = (z_1, \ldots, z_S)$, where each $z_s$ is drawn from a learned categorical distribution over a fixed latent vocabulary $\mathcal{V}$. These tokens are embedded and concatenated with the input token embeddings to form a joint sequence processed autoregressively. The architecture is defined by:

$$z_s \sim \text{Categorical}(\pi_\theta(x_{\leq t}, z_{<s})) \quad \forall s \in \{1, \ldots, S\}, \tag{1}$$

$$\mathbf{H}_{\text{input}} = \text{Concat}\big(e(x_1), \ldots, e(x_T), e(z_1), \ldots, e(z_S)\big), \tag{2}$$

where $e(\cdot)$ denotes the token embedding function and $\pi_\theta$ is a parameterized policy conditioned on prior inputs and latent tokens. This formulation enables direct interpretability and intervention at

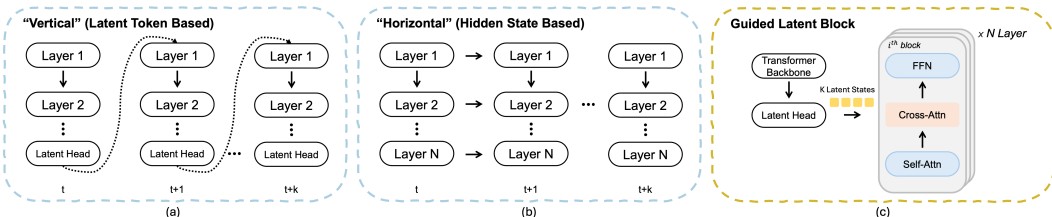

Figure 2: Architectural comparison of latent representation paradigms. (a) *Vertical*: Discrete tokens processed autoregressively. (b) *Horizontal*: Continuous hidden states injected into residual stream. (c) *Hybrid (Ours)*: Continuous latent states as per-layer guidance via cross-attention.

the token level. However, the information capacity of $Z$ is limited by the discrete tokens, typically ~15 bits per token (Zhu et al., 2025), restricting the complexity of representable reasoning states and undermining the expressive potential of continuous latent spaces.

**Horizontal Paradigm: Continuous Hidden States**   The horizontal paradigm identifies $Z$ with a subset of the model's internal continuous hidden states. Specifically, $Z$ is extracted from the transformer layer and re-injected into subsequent layers. The architecture is formalized as:

$$Z = h_t^{(l)} \in \mathbb{R}^d, \tag{3}$$

$$\mathbf{H}_{\text{input}}^{(l+1)} = \text{Concat}\big(\mathbf{H}^{(l)}, Z\big), \tag{4}$$

where $h_t^{(l)}$ is the hidden state at layer $l$ and position $t$, and $\mathbf{H}^{(l)} \in \mathbb{R}^{T \times d}$ denotes the full sequence of activations at that layer. This approach preserves high information bandwidth—each $d$-dimensional vector encodes $O(d)$ bits—but entangles reasoning states with linguistic representations. As a result, the model struggles to disentangle task-agnostic reasoning dynamics from surface-level language features, complicating regularization, interpretation, and cross-task generalization.

**Hybrid Paradigm: Guided Latent Reasoning**   We propose **guided latent reasoning** (Figure 2 (c)), a novel architectural paradigm that addresses the limitations of both vertical and horizontal approaches by decoupling the latent reasoning state from the autoregressive token stream while enabling fine-grained, layer-specific influence. This design preserves the structural clarity of discrete tokens while leveraging the representational capacity of continuous hidden states.

The key innovation treats the latent variable $Z = \{Z_1, \ldots, Z_K\}$ as an external guidance bank $\mathbf{Z} \in \mathbb{R}^{K \times d}$ that provides global contextual guidance. Inspired by conditioning mechanisms in Diffusion Transformers Peebles & Xie (2023), $\mathbf{Z}$ is sampled once during training or inference and shared across all transformer layers:

$$\mathbf{Z} = \text{MLP}_{\text{latent}}\big([\mathbf{H}^{\text{backbone}}]\big) \in \mathbb{R}^{K \times d},$$

where $\mathbf{H}^{\text{backbone}}$ is obtained from the backbone transformer processing the input context.

Rather than interleaving $Z$ with tokens or overwriting activations, we augment each transformer block with cross-attention where $\mathbf{Z}$ serves as query and the self-attended representations provide keys and values:

$$\mathbf{H}_{\text{self}}^{(l)} = \text{SelfAttn}\left(\text{LayerNorm}(\mathbf{H}^{(l-1)})\right) + \mathbf{H}^{(l-1)}, \tag{5}$$

$$\mathbf{H}_{\text{cross}}^{(l)} = \text{CrossAttn}\left(\text{LayerNorm}(\mathbf{Z}), \text{LayerNorm}(\mathbf{H}_{\text{self}}^{(l)}), \text{LayerNorm}(\mathbf{H}_{\text{self}}^{(l)})\right), \tag{6}$$

$$\mathbf{H}_{\text{merged}}^{(l)} = \mathbf{H}_{\text{self}}^{(l)} + g_l \cdot \mathbf{H}_{\text{cross}}^{(l)}, \tag{7}$$

where $g_l$ is a learnable gate that modulates guidance strength per layer.

This establishes a clean separation between the *reasoning trace* (evolving token representations) and *reasoning state* (external $\mathbf{Z}$). The adaptive gating $g_l$ naturally aligns with transformer layer specialization—minimizing interference in shallow layers while amplifying reasoning influence in deeper layers Geva et al. (2020). Critically, since $\mathbf{Z}$ resides outside the token sequence, it preserves full autoregressive compatibility without consuming sequence length or disrupting causal masking. This hybrid approach achieves an optimal balance: maintaining the expressive power of continuous latent spaces while providing precise architectural control over reasoning dynamics.

## 4 EXPERIMENTS

Table 1: Main results on mathematical and commonsense reasoning benchmarks. We compare our **variCoT** variants against strong baselines across two model families. The best score for each dataset is in **bold**. The best score among our proposed variants is underlined.

| Model | GSM8k | GSM8k-NL | CommonsenseQA | SVAMP | GSM-Hard | MultiA |
|---|---|---|---|---|---|---|
| **GPT-2** | | | | | | |
| CoT-SFT | **44.1** | 34.8 | 36.9 | 41.8 | 9.8 | 90.7 |
| No-CoT-SFT | 19.1 | 19.1 | 20.5 | 16.4 | 4.3 | 41.1 |
| Pause-CoT-SFT | 16.4 | 16.4 | - | 14.8 | 4.1 | 39.2 |
| iCoT | 30.1 | 3.2 | 26.2 | 29.4 | 5.7 | 55.5 |
| Coconut | 34.1 | 24.9 | 38.6 | 36.4 | 7.9 | 82.2 |
| CODI | 43.7 | 35.3 | **44.0** | **42.9** | **9.9** | **92.8** |
| variCoT-Vertical | 38.5 | 31.5 | 38.0 | 37.0 | 9.2 | 84.8 |
| variCoT-Horizontal | 39.6 | 32.0 | 37.3 | 37.8 | 9.4 | 83.6 |
| variCoT-Guided | 43.9 | **35.4** | 37.9 | 42.6 | 9.4 | 91.5 |
| **LLaMA3.2-1b** | | | | | | |
| CoT-SFT | **61.6** | **54.1** | 68.2 | **66.7** | **15.8** | **99.3** |
| No-CoT-SFT | 30.9 | 30.9 | 74.9 | 44.1 | 7.1 | 70.9 |
| Pause-CoT-SFT | 28.1 | 28.1 | - | 41.2 | 6.7 | 65.3 |
| iCoT | 19.0 | 15.2 | 72.6 | 40.9 | 4.4 | 39.0 |
| Coconut | 45.3 | 27.2 | 60.6 | 48.8 | 9.9 | 90.1 |
| CODI | 55.6 | 49.7 | 74.0 | 61.1 | 12.8 | 96.1 |
| variCoT-Vertical | 51.3 | 42.8 | 77.2 | 61.3 | 13.3 | 94.1 |
| variCoT-Horizontal | 51.5 | 43.0 | 76.4 | 60.8 | 13.1 | 94.3 |
| variCoT-Guided | 57.5 | 53.75 | **78.1** | 65.2 | 15.6 | 98.5 |

We conducted experiments on both GPT2 Radford et al. (2019) and LLaMA3.2-1b Grattafiori et al. (2024) to validate the generalizability of our method across different foundation models. For training, we employed the AdamW (Loshchilov & Hutter, 2017) optimizer with a learning rate of $5 \times 10^{-5}$, incorporating 10% warm-up steps followed by linear decay. The GPT2 model (Radford et al., 2019) was trained for 30 epochs, while LLaMA3.2-1b (Grattafiori et al., 2024) was trained for 15 epochs, both with an effective batch size of 256. Regarding hyperparameter configuration, we selected 6 latent reasoning embeddings with $\beta = 0.01$ to align with other methods in the baseline; further hyperparameter analysis can be found in our ablation studies. To ensure reproducibility, we set a fixed random seed (seed=42) for all experiments, and each reported result represents a single run under this controlled setting. All experiments were performed on an ml.p5en.48xlarge instance of Amazon Elastic Compute Cloud, which includes 8 NVIDIA H200 (141GB) GPUs, using PyTorch 2.6 (Paszke et al., 2019) as the deep learning framework.

**Dataset** Following Shen et al. (2025), we evaluate **variCoT** on six public datasets, categorized into in-domain and out-of-domain (OOD) settings for evaluation. We use three datasets for in-domain evaluation. **GSM8k-Aug** (Deng et al., 2023) is a math reasoning dataset of 385K samples, augmented from GSM8K (Cobbe et al., 2021) using GPT-4, with structured mathematical expressions as rationales. **GSM8k-Aug-NL** Shen et al. (2025) is a variant of GSM8k-Aug where the reasoning process is presented in natural language. **CommonsenseQA-CoT** (Shen et al., 2025), which extends the original CommonsenseQA (Talmor et al., 2018) with Chain-of-Thought (CoT) annotations that were generated using GPT-4o-mini and filtered for correctness. To evaluate robustness, we train on GSM8k-Aug and test on three OOD datasets. **SVAMP** (Patel et al., 2021) is an elementary school math word problem dataset. **GSM-HARD** (Gao et al., 2023) is a more challenging version of the GSM8K test set with an expanded value range. **MultiArith** (Roy & Roth, 2015) is a multi-step arithmetic word problem dataset from MAWPS (Koncel-Kedziorski et al., 2016).

**Baselines** We compare our method, **variCoT**, against several strong baselines that explore explicit and implicit reasoning: **CoT-SFT**, standard supervised fine-tuning (SFT) on explicit chain-of-thought

demonstrations, where the model generates the reasoning process before the final answer at inference; **No-CoT-SFT**, standard SFT on question-answer pairs only, without explicit reasoning steps; **Pause-CoT-SFT** (Goyal et al., 2024), SFT with special `<pause>` tokens inserted before the answer to encourage implicit reasoning (we use 6 for a fair comparison); **iCoT** (Deng et al., 2024), a strategy that internalizes reasoning by gradually removing the explicit CoT during training to ultimately output only the final answer; and **COCONUT** (Hao et al., 2024), a method that also internalizes the CoT, but replaces it with learned implicit reasoning tokens instead of deleting it. **CODI** (Shen et al., 2025), a method that also internalizes the CoT, but uses a distillation framework to compress the knowledge from an explicit CoT (teacher) process into a series of continuous thought tokens (student).

## 4.1 MAIN RESULTS

Table 1 presents comprehensive evaluations across mathematical and commonsense reasoning benchmarks. Our variCoT framework demonstrates strong performance across both GPT-2 and LLaMA3.2-1B model families, consistently matching or exceeding the accuracy of explicit CoT-SFT while offering significant efficiency gains.

On GPT-2, variCoT achieves 43.9% accuracy on GSM8K and 91.5% on MultiArith, performing competitively with explicit CoT-SFT (44.1% and 90.7% respectively) while significantly outperforming other implicit reasoning methods. The framework shows particular strength on out-of-domain generalization, achieving 42.6% on SVAMP and 9.4% on GSM-HARD, demonstrating robust reasoning capabilities without explicit intermediate token generation.

The performance advantage scales effectively to the larger LLaMA3.2-1B model, where variCoT achieves 57.5% on GSM8K and 98.5% on MultiArith—closely approaching CoT-SFT performance (61.6% and 99.3%) while offering the efficiency benefits of latent reasoning. Notably, our method shows superior commonsense reasoning capabilities, achieving 78.1% on CommonsenseQA-CoT, outperforming all baselines including explicit CoT-SFT (68.2%).

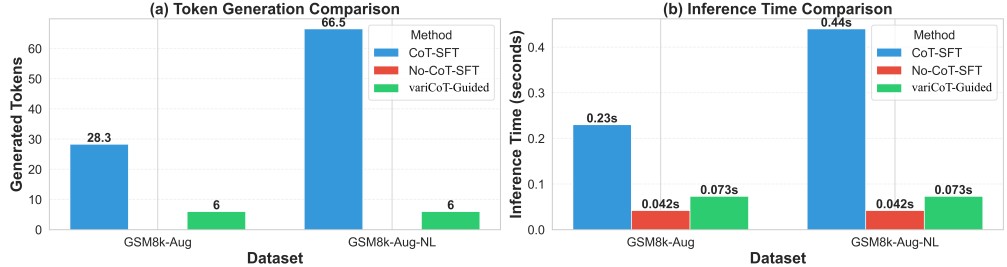

Figure 3: Inference efficiency of different methods in different datasets. The left side shows the average number of CoTs generated during the inference process, and the right side shows the average duration of complete inference, with GPT-2 Small as the base model.

In terms of inference efficiency, variCoT demonstrates significant advantages. As shown in Figure 3, our method reduces token generation by approximately 80-90% compared to CoT-SFT, requiring only 6 latent tokens instead of lengthy reasoning chains. This translates to a 70-80% reduction in inference time (0.073s vs. 0.32s for CoT-SFT on GSM8K), while maintaining competitive accuracy. Although slightly slower than No-CoT-SFT, this small efficiency sacrifice is exchanged for substantial performance gains, providing an excellent balance between efficiency and reasoning capability.

Table 2: CoT Reconstruction Quality Evaluation

| Model | GSM8K-Aug | | GSM8K-NL-Aug | |
|---|---|---|---|---|
| | ROUGE-1 | BLEU-1 | ROUGE-1 | BLEU-1 |
| GPT-2 | 0.69 | 0.66 | 0.63 | 0.62 |
| LLaMA-1B | 0.78 | 0.72 | 0.72 | 0.69 |

A key advantage of variCoT is its reversible reasoning capability. As shown in Table 2, our model achieves high reconstruction fidelity with ROUGE-1 scores of 0.69 (GPT-2) and 0.78 (LLaMA-1B) on GSM8K, indicating that the latent embeddings effectively capture essential reasoning information.

This provides significant interpretability advantages over other implicit CoT methods, as demonstrated by the reconstruction examples in Figure 7.

## 4.2 ABLATION STUDIES

We conduct systematic ablations to understand the impact of key architectural choices and hyperparameters. First, we compare the three latent representation paradigms introduced in Section 3.2. The guided latent reasoning approach consistently outperforms both vertical (discrete token) and horizontal (continuous hidden state) variants across all benchmarks. On GPT-2, the guided paradigm achieves 43.9% on GSM8K compared to 38.5% for vertical and 39.6% for horizontal approaches. This advantage is even more pronounced on LLaMA3.2-1B, where the guided approach reaches 57.5% versus 51.3% and 51.5% for the alternatives. The results validate our architectural design that decouples reasoning guidance from linguistic processing through cross-attention mechanisms.

Figure 6 shows the sensitivity analysis of the number of latent reasoning embeddings. We find optimal performance scales with number of latent reasoning embeddings, consistent with findings in prior work Zhu et al. (2025). This configuration balances representational capacity with training stability, providing sufficient bandwidth for complex reasoning while avoiding overfitting. Performance degrades with fewer embeddings due to limited expressivity, while excessive embeddings introduce noise and optimization challenges.

## 5 RELATED WORK

Chain-of-Thought (CoT) prompting has established a powerful paradigm for multi-step reasoning in large language models by verbalizing intermediate steps (Wei et al., 2022). However, constraining reasoning to a discrete token space introduces significant latency and limits expressive power, motivating a shift towards *implicit* or *latent* CoT, where reasoning occurs in the model's continuous hidden states (Zhu et al., 2025). Current latent reasoning methods primarily fall into two categories: *vertical* approaches that increase effective model depth by iteratively refining activations within a fixed set of layers (Geiping et al., 2025; Mohtashami et al., 2023), and *horizontal* approaches that expand temporal context by propagating compressed hidden states over time (Dao & Gu, 2024; Behrouz et al., 2024). While these methods enhance reasoning, they often require specialized architectures or entangle reasoning states with linguistic representations.

A parallel line of work induces latent reasoning capabilities through specialized training objectives on standard Transformer architectures. These strategies include using special pause tokens to encourage implicit computation (Goyal et al., 2024), progressively internalizing explicit CoT steps during fine-tuning (Deng et al., 2024), or compressing natural language rationales into continuous thought vectors via knowledge distillation (Hao et al., 2024; Shen et al., 2025). Although effective, these training-induced methods often depend on multi-stage pipelines or heuristic objectives, lacking a unified, end-to-end optimization framework.

Variational inference, while a cornerstone of generative modeling, remains nascent in the context of continuous latent reasoning. Prior works have either relied on discrete latent tokens (Su et al., 2025) or learned compressed reasoning traces without a principled probabilistic foundation (Zhang et al., 2025), failing to provide a robust framework for structured stochastic inference.

Our work variCoT, addresses these gaps by proposing a unified variational framework that formalizes latent reasoning as principled stochastic inference. Unlike prior methods, variCoT is optimized end-to-end via a single, theoretically grounded evidence lower bound (ELBO) objective within a standard Transformer. It introduces a *guided latent reasoning* mechanism that synthesizes the benefits of vertical depth and horizontal recurrence, using cross-attention to decouple abstract reasoning from its linguistic realization. This unique design enables efficient, latent-only inference for fast decoding while preserving the ability to generate explicit CoT for interpretability—a critical capability not offered by previous implicit reasoning methods.

## 6 CONCLUSION

Chain-of-Thought reasoning improves LLM performance but incurs significant computational overhead through sequential token generation. Existing implicit CoT methods rely on heuristic archi-

tectures and multi-stage training, lacking principled optimization. We introduce variCoT, a unified variational framework that overcomes these limitations through an evidence lower bound objective, formalizing latent reasoning traces as continuous stochastic variables.

Our framework combines strategic control tokens for end-to-end training with guided latent reasoning that decouples abstract computation from linguistic realization. Experiments show variCoT matches or exceeds explicit CoT accuracy while providing $2.5\times$ faster inference and reversible reasoning capability. This establishes a theoretically grounded, scalable approach to efficient reasoning that bridges continuous latent spaces with autoregressive generation.

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

## A    APPENDIX

### A.1    PROOF OF THEOREM 1: EVIDENCE LOWER BOUND FOR VARICOT

*Proof.* We begin by deriving the ELBO for the marginal log-likelihood $\log p(Y^r, Y^a \mid X^q)$. Introducing the variational posterior $q_\phi(Z \mid X^q, Y^r, Y^a)$, we have:

$$\log p(Y^r, Y^a \mid X^q) = \log \int p(Y^r, Y^a, Z \mid X^q) dZ$$

$$= \log \int q_\phi(Z \mid X^q, Y^r, Y^a) \frac{p(Y^r, Y^a, Z \mid X^q)}{q_\phi(Z \mid X^q, Y^r, Y^a)} dZ$$

$$\geq \mathbb{E}_{q_\phi}\left[\log \frac{p(Y^r, Y^a, Z \mid X^q)}{q_\phi(Z \mid X^q, Y^r, Y^a)}\right] \quad \text{(Jensen's inequality)}$$

$$= \mathbb{E}_{q_\phi}\left[\log p(Y^r, Y^a, Z \mid X^q) - \log q_\phi(Z \mid X^q, Y^r, Y^a)\right].$$

Let $\mathcal{L}_{\text{ELBO}} = \mathbb{E}_{q_\phi}\left[\log \frac{p(Y^r, Y^a, Z \mid X^q)}{q_\phi(Z \mid X^q, Y^r, Y^a)}\right]$. We can rewrite the marginal likelihood as:

$$\log p(Y^r, Y^a \mid X^q) = \mathbb{E}_{q_\phi}\left[\log p(Y^r, Y^a \mid X^q)\right]$$

$$= \mathbb{E}_{q_\phi}\left[\log \frac{p(Y^r, Y^a, Z \mid X^q)}{p(Z \mid X^q, Y^r, Y^a)}\right]$$

$$= \mathbb{E}_{q_\phi}\left[\log \frac{p(Y^r, Y^a, Z \mid X^q)}{q_\phi(Z \mid X^q, Y^r, Y^a)} \cdot \frac{q_\phi(Z \mid X^q, Y^r, Y^a)}{p(Z \mid X^q, Y^r, Y^a)}\right]$$

$$= \mathcal{L}_{\text{ELBO}} + D_{\text{KL}}\left(q_\phi(Z \mid X^q, Y^r, Y^a) \,\|\, p(Z \mid X^q, Y^r, Y^a)\right).$$

Since the KL divergence is non-negative, we have $\log p(Y^r, Y^a \mid X^q) \geq \mathcal{L}_{\text{ELBO}}$, with equality if and only if $q_\phi(Z \mid X^q, Y^r, Y^a) = p(Z \mid X^q, Y^r, Y^a)$. $\qquad\square$

### A.2    PROOF OF THEOREM 2: VARICOT OBJECTIVE DECOMPOSITION

*Proof.* Starting from the ELBO expression in Theorem 1:

$$\mathcal{L}_{\text{ELBO}} = \mathbb{E}_{q_\phi}\left[\log \frac{p(Y^r, Y^a, Z \mid X^q)}{q_\phi(Z \mid X^q, Y^r, Y^a)}\right],$$

we substitute the joint distribution from Propositon 2.3:

$$\mathcal{L}_{\text{ELBO}} = \mathbb{E}_{q_\phi}\left[\log \frac{p_\psi(Y^r \mid X^q, Z) \cdot p_\rho(Y^a \mid X^q, Z) \cdot p_\theta(Z \mid X^q)}{q_\phi(Z \mid X^q, Y^r, Y^a)}\right]$$

$$= \mathbb{E}_{q_\phi}\left[\log p_\psi(Y^r \mid X^q, Z) + \log p_\rho(Y^a \mid X^q, Z) + \log p_\theta(Z \mid X^q) - \log q_\phi(Z \mid X^q, Y^r, Y^a)\right].$$

By linearity of expectation:

$$\mathcal{L}_{\text{ELBO}} = \mathbb{E}_{q_\phi}\left[\log p_\psi(Y^r \mid X^q, Z)\right] + \mathbb{E}_{q_\phi}\left[\log p_\rho(Y^a \mid X^q, Z)\right]$$

$$+ \mathbb{E}_{q_\phi}\left[\log \frac{p_\theta(Z \mid X^q)}{q_\phi(Z \mid X^q, Y^r, Y^a)}\right].$$

The third term can be rewritten as a KL divergence:

$$\mathbb{E}_{q_\phi}\left[\log \frac{p_\theta(Z \mid X^q)}{q_\phi(Z \mid X^q, Y^r, Y^a)}\right] = -D_{\text{KL}}\left(q_\phi(Z \mid X^q, Y^r, Y^a) \,\|\, p_\theta(Z \mid X^q)\right).$$

Thus, we obtain the final decomposition:

$$\mathcal{L}_{\text{ELBO}} = \mathbb{E}_{q_\phi}\left[\log p_\psi(Y^r \mid X^q, Z)\right] + \mathbb{E}_{q_\phi}\left[\log p_\rho(Y^a \mid X^q, Z)\right]$$

$$- D_{\text{KL}}\left(q_\phi(Z \mid X^q, Y^r, Y^a) \,\|\, p_\theta(Z \mid X^q)\right).$$

The introduction of the $\beta$ coefficient follows the $\beta$-VAE framework to control the strength of the KL regularization term, giving us the final objective:

$$\mathcal{L}_{\text{ELBO}} = \underbrace{\mathbb{E}_{q_\phi}\left[\log p_\psi(Y^r \mid X^q, Z)\right]}_{\mathcal{L}_{\text{reasoning}}} + \underbrace{\mathbb{E}_{q_\phi}\left[\log p_\rho(Y^a \mid X^q, Z)\right]}_{\mathcal{L}_{\text{answer}}}$$

$$- \underbrace{\beta \cdot D_{\text{KL}}\left(q_\phi(Z \mid X^q, Y^r, Y^a) \,\|\, p_\theta(Z \mid X^q)\right)}_{\mathcal{L}_{\text{KL}}}.$$

$\square$

## A.3 STRATEGIC CONTROL TOKENS AND END-TO-END TRAINING PROTOCOL

To ensure full reproducibility and clarity, we detail the complete token-level specification of `variCoT`'s control mechanism, including exact input formats for all components and the inference procedure.

**Control Token Specification**  We define four functionally specialized control tokens that orchestrate variational inference within a single Transformer pass. All components share the same set of parameters, and gradients flow end-to-end through the entire sequence.

- **Prior Network** $p_\theta(Z \mid X^q)$:
  Input sequence: `<prompt>` $X^q$ `</prompt>` `<prior>`
  The latent variable $Z \in \mathbb{R}^{K \times d}$ is sampled from a distribution parameterized by the hidden state at the `<prior>` token position. During inference, this is the sole source of $Z$.

- **Posterior Network** $q_\phi(Z \mid X^q, Y^r, Y^a)$:
  Input sequence: `<prompt>` $X^q$ `</prompt>` `<cot>` $Y^r$ `<eos>` `<answer>` $Y^a$ `<eos>` `<posterior>`
  The approximate posterior is inferred from the hidden state at `<posterior>`, conditioned on both the ground-truth reasoning chain $Y^r$ and final answer $Y^a$. During training, $Z$ is sampled from this posterior.

- **Reasoning Decoder** $p_\psi(Y^r \mid X^q, Z)$:
  Input sequence: `<prompt>` $X^q$ `</prompt>` `<latent>` $Z_1, \ldots, Z_K$ `</latent>` `<cot>`
  Target sequence: $Y^r$ followed by `<eos>`. The latent state $Z$ is injected via the Guided Latent Reasoning mechanism (Section 3.2). During training, $Z$ is sampled from the posterior; during inference, from the prior.

- **Answer Decoder** $p_\rho(Y^a \mid X^q, Z)$:
  Input sequence: `<prompt>` $X^q$ `</prompt>` `<latent>` $Z_1, \ldots, Z_K$ `</latent>` `<answer>`
  Target sequence: $Y^a$ followed by `<eos>`.

The `<latent>` token sequence serves as a placeholder that triggers the latent injection mechanism; its embeddings are unused—the actual latent vectors **Z** are provided externally via cross-attention (Section 3.2).

**Training Protocol**  During training, we construct a single concatenated sequence:

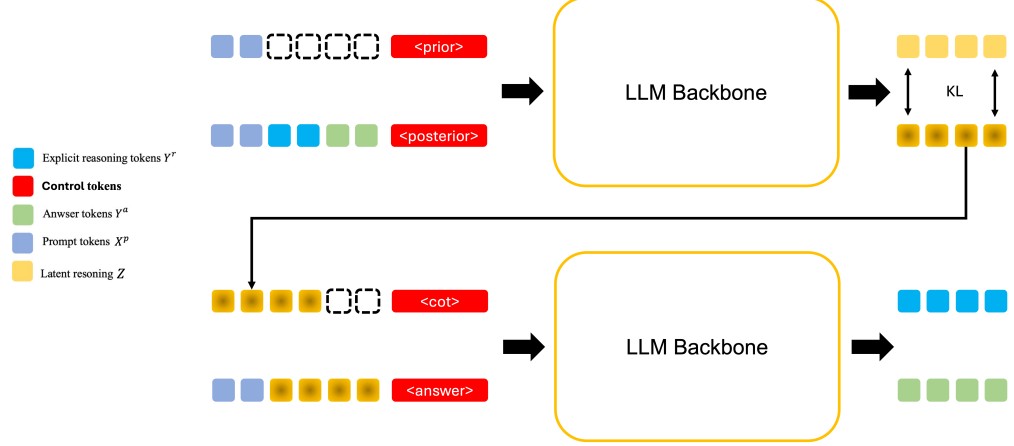

Figure 4: Training data flow in `variCoT`. Control tokens (`<prior>`, `<posterior>`, `<cot>`, `<answer>`) condition distinct probabilistic operations within a single forward pass. The latent variable $Z$ is sampled from the posterior and routed to decoders via the `<latent>` token sequence. Parameter sharing enables end-to-end gradient flow.

$$\texttt{<prompt>}\ X^q\ \texttt{</prompt>}\ \texttt{<cot>}\ Y^r\ \texttt{<eos>}\ \texttt{<answer>}\ Y^a\ \texttt{<eos>}\ \texttt{<posterior>}\ \texttt{<latent>}$$
$$Z_1, \ldots, Z_K\ \texttt{</latent>}$$

The model first processes the context up to `<posterior>` to infer $q_\phi(Z \mid X^q, Y^r, Y^a)$, samples $Z$, and then uses this $Z$ to condition the subsequent generation of $Y^r$ and $Y^a$. The entire sequence is trained with standard autoregressive language modeling loss, enabling end-to-end optimization.

**Inference Protocol** At inference time, no ground-truth reasoning trace or answer is available. The model proceeds in two stages: 1. **Latent sampling**: Process <prompt> $X^q$ </prompt> <prior> to obtain

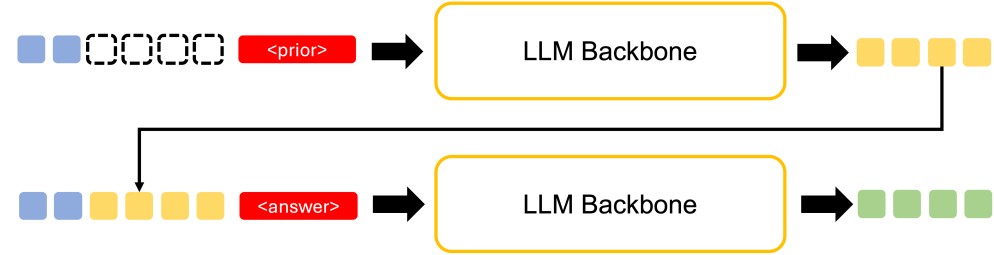

Figure 5: Inference in variCoT. The prior network samples $Z$ from $p_\theta(Z \mid X^q)$. The same latent state is then used to generate $Y^r$ and $Y^a$ sequentially. No ground-truth reasoning traces are required.

$p_\theta(Z \mid X^q)$, then sample $Z$. 2. **Autoregressive generation**: Using the sampled $Z$, generate either: - A reasoning chain: <prompt> $X^q$ </prompt> <latent> $Z$ </latent> <cot> $\rightsquigarrow Y^r$ - A direct answer: <prompt> $X^q$ </prompt> <latent> $Z$ </latent> <answer> $\rightsquigarrow Y^a$

Both generations are fully autoregressive and leverage the same latent state $Z$, enabling coherent, uncertainty-aware predictions.

This unified design eliminates the need for external encoders, distillation teachers, or non-autoregressive memory modules, while preserving full compatibility with standard transformer-based language models.

## A.4  MORE EXPERIMENT RESULTS

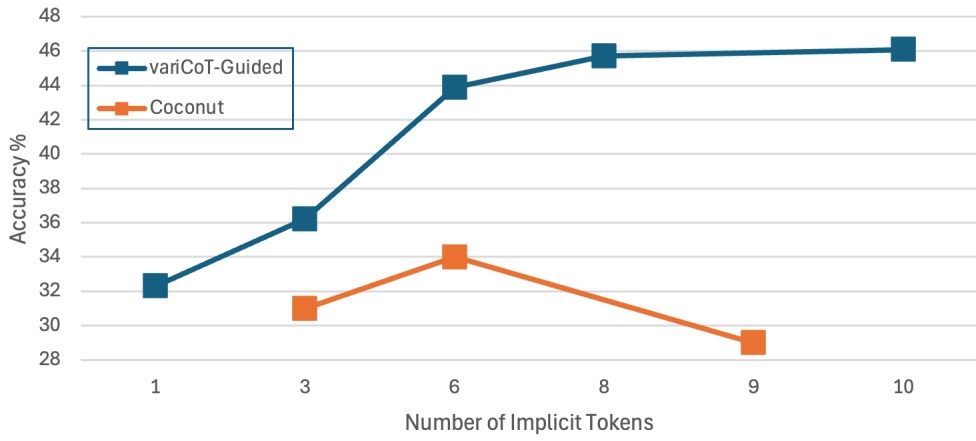

Figure 6: The impact of different numbers of implicit reasoning embeddings and different lambda settings on performance under the GSM8K-Aug dataset, with GPT-2 Small as the base model.

## A.5 EXPLICIT REASONING RECONSTRUCTION VISUALIZATION

Example 1

> **Question:**
> Josh decides to try flipping a house. He buys a house for $80,000 and then puts in $50,000 in repairs. This increased the value of the house by 150%. How much profit did he make?
> **Original Reasoning:**
> <<80000+50000=130000>> <<80000*1.5=120000>> <<120000+80000=200000>> <<200000-130000=70000>>
> **Reversible Latent Decoder Output:**
> <<80000+50000=130000>> <<150%*80000=120000>> <<80000+120000=200000>> <<200000-130000=70000>>

Example 2

> **Question:**
> Janet's ducks lay 16 eggs per day. She eats three for breakfast every morning and bakes muffins for her friends every day with four. She sells the remainder at the farmers' market daily for $2 per fresh duck egg. How much in dollars does she make every day at the farmers' market?
> **Original Reasoning:**
> Janet sells 16 - 3 - 4 = 9 duck eggs a day. She makes 9 * 2 = $18 every day at the farmer's market.
> **Reversible Latent Decoder Output:**
> Janet has 16 eggs daily, leaving 16 - 7 = 9 eggs to sell. At $2 per egg, she earns 9 × $2 = $18.

Figure 7: Reconstructed performance of implicit reasoning.

How to interpret implicit reasoning has been a key challenge in this direction, especially for scenarios that require explicit reasoning generation, such as mathematical proofs, logical attributions, etc. Although pure implicit reasoning can achieve efficiency improvements, it cannot effectively obtain explicit reasoning processes. Thanks to the *Reversible Latent Decoder*, our proposed variCoT can directly reconstruct explicit reasoning based on implicit reasoning, which offers better interpretability compared to other implicit reasoning methods. As shown in Fig. 7, implicit reasoning embeddings can be directly reconstructed into explicit reasoning by the *Reversible Latent Decoder*, and can also support the generation of final answers in terms of thought paths, which more intuitively demonstrates the superiority of our proposed method.

## A.6 TRAINING AND INFERENCE ALGORITHMS

---

**Algorithm 1** variCoT Training Procedure

---

**Require:** Dataset $\mathcal{D} = \{(X^q, Y^r, Y^a)\}$, model parameters $\theta, \phi, \psi, \rho$

1: **while** not converged **do**
2:      Sample batch $(X^q, Y^r, Y^a) \sim \mathcal{D}$
3:      Construct input sequence: $S = [X^q, \texttt{<cot>}, Y^r, \texttt{<eos>}, \texttt{<answer>}, Y^a, \texttt{<eos>}, \texttt{<posterior>}]$
4:      Compute hidden states: $\mathbf{H} = \text{Transformer}(S)$
5:      Extract posterior parameters from $\mathbf{h}_{\texttt{<posterior>}}$: $\mu_\phi, \sigma_\phi$
6:      Sample latent: $Z \sim \mathcal{N}(\mu_\phi, \sigma_\phi^2)$
7:      Construct decoding sequence: $S_{\text{dec}} = [X^q, \texttt{<latent>}, Z, \texttt{<cot>}]$
8:      Compute reasoning loss: $\mathcal{L}_{\text{reasoning}} = -\log p_\psi(Y^r \mid X^q, Z)$
9:      Construct answer sequence: $S_{\text{ans}} = [X^q, \texttt{<latent>}, Z, \texttt{<answer>}]$
10:      Compute answer loss: $\mathcal{L}_{\text{answer}} = -\log p_\rho(Y^a \mid X^q, Z)$
11:      Compute KL divergence: $\mathcal{L}_{\text{KL}} = D_{\text{KL}}(q_\phi(Z \mid X^q, Y^r, Y^a) \| p_\theta(Z \mid X^q))$
12:      Total loss: $\mathcal{L} = \mathcal{L}_{\text{reasoning}} + \mathcal{L}_{\text{answer}} + \beta \mathcal{L}_{\text{KL}}$
13:      Update parameters via gradient descent: $\nabla_\theta \mathcal{L}$
14: **end while**

---

**Algorithm 2** variCoT Inference Procedure

**Require:** Input question $X^q$, trained model parameters $\theta, \rho$
1: Construct prior sequence: $S_{\text{prior}} = [X^q, \texttt{<prior>}]$
2: Compute hidden states: $\mathbf{H} = \text{Transformer}(S_{\text{prior}})$
3: Extract prior parameters from $\mathbf{h}_{\texttt{<prior>}}$: $\mu_\theta, \sigma_\theta$
4: Sample latent: $Z \sim \mathcal{N}(\mu_\theta, \sigma_\theta^2)$
5: Construct answer sequence: $S_{\text{ans}} = [X^q, \texttt{<latent>}, Z, \texttt{<answer>}]$
6: Generate answer autoregressively: $Y^a \sim p_\rho(\cdot \mid X^q, Z)$
7: **Optional**: Generate reasoning chain: $Y^r \sim p_\psi(\cdot \mid X^q, Z)$
**Ensure:** Final answer $Y^a$ (and optional reasoning chain $Y^r$)

