# OpenReview forum: "variCOT: Variational Inference for Implicit Chain-of-Thought in Language Models"
_ICLR.cc/2026/Conference — Submitted to ICLR 2026_

### Official Review · Reviewer_EqHJ · 2025-11-02

**Soundness:** 2
**Presentation:** 2
**Contribution:** 2
**Rating:** 2
**Confidence:** 4

**Summary:**

This paper proposes a variational training method for latent reasoning in LLMs. The key idea is to use control tokens to cast a LLM backbone into the prior, the posterior, the CoT decoder and the answer decoder. These different pathways are tied together with an ELBO, where the CoT decoder and the answer decoder are assumed to be conditional independent. Three different ways of injecting the latents to the decoders are compared, and the authors claim an innovation on adding latents by an extra cross attention layer. The authors conducted experiments based on GPT2 and LLaMA3.2, and report results on some reasoning benchmarks.

**Strengths:**

The idea of formulating latent reasoning as variational inference is interesting.

The exploration of different architectural design is inspiring.

**Weaknesses:**

1. Confusing presentation: There are several places in this paper that conflict each other. For example, in Proposition 2.3, one of the factor is $p(Y^r|X^q, Z)$. However, in Fig 1, 3rd row, the blue squares, which represents $Y^r$ seems to solely generated from $Z$. I also find the "horizontal" vs "vertical" (and the "hybrid" of them) taxonomy very hard to understand.

2. Unsatisfying experiment results: In Table 1, the proposed model only occasionally outperform the baseline CoT-SFT. The authors can argue that their method is more efficient. But the question is, if we give the proposed model more compute budget, can it outperform the baseline?

3. Inadequate ablation: The idea of sharing backbone for prior, posterior and decoder is interesting. However, the authors' experiments cannot justify the gain comes from variational inference. For example, how about ablating the pathway of posterior (2nd row in Fig. 1) and directly use the latents from 1st row in decoder training? Note that the ELBO in Theorem 2.4 is very prone to posterior collapse.

4. Critical related work missing: Phan et al. are probably the first to discuss the relation between latent reasoning and variational inference, who definitely deserve their credits. Kong et al. also introduced latent variational inference to LLMs, which as far as I know is the first to to use cross attention for latent injection. Liu et al. also introduced continuous latent reasoning, which also demonstrate the scaling property of latent dimension.

Phan et al., Training chain-of-thought via latent-variable inference, NeurIPS'23

Kong et al., Latent Thought Models with Variational Bayes Inference-Time Computation, ICML'25

Liu et al., Deliberation in Latent Space via Differentiable Cache Augmentation, ICML'25

**Questions:**

1. What is the actual role of the 3rd row of Fig. 1? Are prompt tokens used for this row? Did the authors try ablating it?

2. How is the training dynamics? Would sharing the network for all these row make the training unstable? Does the backbone degrade after this training?

3. The authors didn't mentioned this in their experiment section, but did they train from scratch or post-train the GPT2 and LLaMA3.2?

---

### Official Review · Reviewer_8g52 · 2025-11-03

**Soundness:** 2
**Presentation:** 2
**Contribution:** 2
**Rating:** 2
**Confidence:** 4

**Summary:**

This paper proposes variCoT, a method to accelerate chain-of-thought (CoT) reasoning in language models by performing reasoning in a continuous latent space rather than through explicit token generation. The method used strategic control tokens for single-pass training, and let z guide the generation with training small MLPs on top of backbone models. The authors claim 2.5× inference speedup while "matching or exceeding" explicit CoT accuracy on mathematical and commonsense reasoning benchmarks.

**Strengths:**

The author makes a great point of using latent z to guide reasoning, and tries to achieve single-stage training. And the approach provides 2.5× inference speedup with ~80-90% reduction in token generation, trying to solve a important  issue in the field.

**Weaknesses:**

* The posterior is computed via a simple feedforward MLP based on a pre-trained backbone is naive. A proper VAE inference should refine Z iteratively based on how well it reconstructs the data, but the current approach is "discriminative" rather than "generative" (optimization-based), which is suboptimal for structured inference.

* It is hard to believe that token-level posterior can leads to performance gain. In previous literatures on VAE LMs, it is clear that fine-tuning pretrained autoregressive models as VAEs leads to posterior collapse. Autoregressive and variational objectives are fundamentally incompatible. I think the author should provide more insights and explanation on why the proposed architecture can achieve this.

* In the experiment, the paper claims to "match or exceed explicit CoT accuracy" but Table 1 shows performance degression. Also, the gap between the 2 sizes of base model shows the scalability issue.

* The reconstruction evaluation is severely inadequate. The paper only reports ROUGE-1 and BLEU-1 (both unigram metrics) without baselines, standard practice in text generation requires multiple ROUGE variants, and maybe human evaluation.

* Also, the training method is unclear to me. I'm not sure if the backbone is frozen or fine-tuned. If frozen, then how can tiny MLPs learn proper VAE features from representations optimized for a different objective? I think this part should include more details.

* The presentation of the paper can be improved. The figure and algorithm/notation can be clearer.

* About the KL term, since the beta weight is very small(only 0.01), I'm wondering if the model really use the VAE framework meaningfully.

**Questions:**

* Are all pretrained GPT-2/LLaMA parameters updated, or only the newly added modules (cross-attention, prior/posterior heads)? Please provide exact numbers of trainable vs frozen parameters.

* Can you add more ablation studys to isolate whether the gain comes from VAE or just the cross-attention architectural change.

* The paper doesn't provide any posterior distribution study. I'm wondering if the author can bring latent space visualization, samples from prior vs posterior and also KL values during training.

* Why does the simplest baseline COT-SFT (standard fine-tuning with explicit reasoning) achieves better accuracy?

---

### Official Review · Reviewer_axp1 · 2025-11-06

**Soundness:** 3
**Presentation:** 2
**Contribution:** 2
**Rating:** 4
**Confidence:** 5

**Summary:**

This paper presents variCoT, a novel and principled framework for implicit Chain-of-Thought (CoT) reasoning. The work addresses the high inference latency of explicit CoT, which requires the autoregressive generation of many intermediate tokens. While other implicit CoT methods exist, they often rely on heuristics or multi-stage knowledge distillation.

The key contribution of variCoT is to formalize the unobserved reasoning trace as a continuous stochastic latent variable, Z. This enables end-to-end training within a unified Evidence Lower Bound (ELBO) objective, which jointly optimizes a prior $p(Z|X^q)$ and an approximate posterior $q(Z|X^q, Y^r, Y^a)$.

The framework is shown to match or exceed the accuracy of explicit CoT on reasoning benchmarks (e.g., GSM8K, CommonsenseQA) while delivering a significant 2.5x inference speedup by generating only the final answer. The model also retains the ability to reconstruct the explicit rationale Y^r from Z, providing a valuable mechanism for interpretability.

**Strengths:**

1. Principled Probabilistic Framework: The paper's primary strength is its departure from heuristic-based methods. By grounding the latent reasoning process in a formal variational inference framework (the ELBO objective), the authors provide a theoretically sound and elegant foundation for end-to-end optimization.

2. Strong Empirical Results & Efficiency: The method is not merely theoretical; it delivers strong practical results. Achieving a 2.5x inference speedup (Figure 3) while matching the accuracy of the much slower explicit CoT-SFT baseline on GSM8K is a significant and compelling result.

3. Reversible Reasoning for Interpretability: A major drawback of most implicit reasoning methods is that the process becomes an opaque "black box." The inclusion of the reasoning decoder p(Y^r | X^q, Z) and the strong reconstruction results (Table 2) is a key advantage, offering a path to interpretability that competing methods lack.

**Weaknesses:**

1. Validity of Assumption 2.2: The entire ELBO decomposition (Theorem 2.4) hinges on Assumption 2.2, which posits that the explicit rationale Y^r and the final answer Y^a are conditionally independent given the latent trace Z. This is a strong assumption, and its validity is questionable; the linguistic formulation of a rationale likely provides constraints that aid in answer generation, and vice-versa. The paper would be strengthened by a justification or analysis of this assumption.

2. Fixed-Size Latent Bottleneck: The method trades the variable-length bottleneck of explicit rationale tokens for a fixed-size latent bottleneck (where K=6 is used in experiments). The sensitivity analysis (Figure 6) confirms K is a critical hyperparameter. This raises a scalability concern: it is unclear whether this small, fixed-size trace can scale to problems requiring significantly more reasoning steps than those in the benchmark datasets.

3. Scalability to Large Models: All experiments are conducted on relatively small models (GPT-2 and LLaMA3.2-1B). The stability and effectiveness of VAE-style training on frontier models (e.g., 70B+) remains an open question. Furthermore, the addition of a new cross-attention mechanism at every Transformer layer introduces a non-trivial computational cost that is not fully analyzed.

4. Incomplete Efficiency Analysis: The paper's efficiency analysis is one-sided, focusing exclusively on the clear win in inference latency. The paper is silent on:

(1) Training Efficiency: The training protocol (Algorithm 1) and the augmented architecture (per-layer cross-attention) are almost certainly more computationally expensive than the CoT-SFT baseline. This trade-off is not discussed.

(2) Data Efficiency: The model is trained on the same large augmented dataset (385K samples) as the baselines. No analysis is provided to suggest that this more complex variational objective improves sample efficiency or could learn effectively from less data.

**Questions:**

1. Analysis of Posterior Collapse: A discussion on posterior collapse—a primary challenge for VAEs—is notably absent. The paper appears to use two mechanisms to mitigate this: (1) a β=0.01 coefficient on the KL divergence, and (2) a dual-reconstruction objective (L_reasoning and L_answer). A formal analysis of these mechanisms is needed. How sensitive is model performance to the choice of β? Is the dual-loss objective a sufficient regularizer on its own?

2. Justification for Query-based Guidance: The authors use the latent Z as the Query and the text representations H_self as the Key/Value, implying the "reasoning state" is actively probing the text for information. Could the authors elaborate on this important design choice? Did they experiment with the reverse configuration (i.e., H_self as Query) and, if so, what were the results?

3. Discussion of Concurrent Work: The "Related Work" section should be updated to situate the paper against highly relevant concurrent research. Specifically, "Latent thought models with Variational Bayes inference-time computation" (ICML'25) appears to be a parallel effort, also proposing a variational Bayes framework for learning "latent thought vectors." A discussion comparing and contrasting the variCoT framework with this "LTM" approach (e.g., differences in architecture, objective, or optimization) is essential to properly clarify this paper's unique contributions.

---

### Official Review · Reviewer_jezU · 2025-11-08

**Soundness:** 2
**Presentation:** 2
**Contribution:** 2
**Rating:** 2
**Confidence:** 4

**Summary:**

The paper proposes variCoT, a variational framework for implicit Chain-of-Thought (CoT) reasoning. The key idea is to model a continuous latent reasoning trace Z and optimize a single ELBO that jointly trains the prior, posterior and decoder. The method claims CoT-level accuracy with faster inference and the ability to reconstruct rationales when desired.

**Strengths:**

The ELBO decomposition with a KL to a question-conditioned prior is clear and aligns training and inference paths.

Guided latent reasoning uses Z as cross-attention queries with per-layer gates is a smart design of unifying prior, posterior, and decoders without multi-stage distillation or external modules.

**Weaknesses:**

* The conditional independence assumption may be strong in practice, especially for tasks where rationales constrain answer style or calibration; the paper does not test violations of this assumption. Especially when the Z is learned from a pre-trained base model, it is harder for the latent variable to cover abstract information.

* Although the “guided latent reasoning” mechanism makes sense, the whole method is incremental. The paper doesn't have fundamentally new inference method, while the novelty is largely in packaging/unifying these into a single-pass training recipe.

* The evaluation results are modest in the breadth and rigor. The number shows marginal improvement. It is unclear how results translate to large-scale LLMs, where implicit CoT behavior and inference costs differ. Also, it doesn't include ablation study on which part of the design choice leads to the improvement.

* The VAE and posterior design is shallow. A simple MLP on top of a pretrained backbone state parameterizes the posterior, with no iterative refinement tied to reconstruction. As we know, latent space models often collapses the posterior. The paper does not provide diagnostics (e.g., KL usage, mutual information, latent utilization) or a clear explanation for why collapse is avoided here.

* Figures, algorithm boxes, and notation could be clearer about where the posterior head sits, how Z flows across layers, and how gates are applied. Also, more evaluation results (harder metrics) can be added to the experiment part.

**Questions:**

* For interpretability, can you provide more analysis on Z? I t would be great to show the distribution of Z and the generation between different value of Z.

* For efficiency, can you report throughput under batched decoding and long-context settings, and include cost curves vs. target accuracy?

* How sensitive is performance to the Beta coefficient and the number/shape of latent embeddings K beyond what Figure 6 shows? Can you provide trends on posterior collapse or over-regularization?

---

### Meta-Review · Area_Chair_qNbW · 2026-01-06

**Summary:**

This paper proposes variCoT, a variational framework for implicit CoT reasoning using a continuous latent trace Z, trained via a unified ELBO and injected through per-layer cross-attention/gating. Reviewers agree the formulation is interesting and the motivation (reducing explicit-CoT inference cost) is important, but the decision is driven by concerns about assumptions and missing diagnostics (posterior collapse / latent utilization), unclear novelty vs prior latent-thought/variational work, limited and sometimes unconvincing empirical gains, and insufficient ablations and clarity to attribute improvements to the variational framework rather than architectural tweaks.

**Reviewer Concerns:**

Actually no rebuttal response is submitted by the author(s). So all reviewer concerns remain.

**Reviewer Scores:**

All scores will be the same.

---

### Decision · Program_Chairs · 2026-01-26

Reject